# Face Recognition at a Distance for a Stand-Alone Access Control System [note 1]

**DOI:** 10.3390/s20030785

**Published:** 2020-01-31

**Authors:** Hansung Lee, So-Hee Park, Jang-Hee Yoo, Se-Hoon Jung, Jun-Ho Huh

**Affiliations:** 1School of Computer Engineering, Youngsan University, 288 Junam-Ro, Yangsan, Gyeongnam 50510, Korea; mohan@ysu.ac.kr; 2Intelligent Convergence Research Laboratory, Electronics and Telecommunications Research Institute (ETRI), 218 Gajeong-ro, Yuseong-gu, Daejeon 34129, Korea; parksh@etri.re.kr; 3Artificial Intelligence Research Laboratory, Electronics and Telecommunications Research Institute (ETRI), 218 Gajeong-ro, Yuseong-gu, Daejeon 34129, Korea; jhy@etri.re.kr; 4School of Major Connection (Bigdata Convergence), Youngsan University, 288 Junam-Ro, Yangsan, Gyeongnam 50510, Korea; 5Department of Data Informatics, Korea Maritime and Ocean University, Busan 49112, Korea

**Keywords:** artificial intelligence, access control, face identification, face recognition at distance, face biometric, face recognition

## Abstract

Although access control based on human face recognition has become popular in consumer applications, it still has several implementation issues before it can realize a stand-alone access control system. Owing to a lack of computational resources, lightweight and computationally efficient face recognition algorithms are required. The conventional access control systems require significant active cooperation from the users despite its non-aggressive nature. The lighting/illumination change is one of the most difficult and challenging problems for human-face-recognition-based access control applications. This paper presents the design and implementation of a user-friendly, stand-alone access control system based on human face recognition at a distance. The local binary pattern (LBP)-AdaBoost framework was employed for face and eyes detection, which is fast and invariant to illumination changes. It can detect faces and eyes of varied sizes at a distance. For fast face recognition with a high accuracy, the Gabor-LBP histogram framework was modified by substituting the Gabor wavelet with Gaussian derivative filters, which reduced the facial feature size by 40% of the Gabor-LBP-based facial features, and was robust to significant illumination changes and complicated backgrounds. The experiments on benchmark datasets produced face recognition accuracies of 97.27% on an E-face dataset and 99.06% on an XM2VTS dataset, respectively. The system achieved a 91.5% true acceptance rate with a 0.28% false acceptance rate and averaged a 5.26 frames/sec processing speed on a newly collected face image and video dataset in an indoor office environment.

## 1. Introduction

With significant advances in biometrics technologies, user authentication systems based on biometrics, such as an iris, fingerprint, and human face, has increased markedly for providing more efficient and secure access control compared to the conventional user authentication system, which uses a password or an Integrated Circuit (IC) card. However, some biometric-based access controls, e.g., fingerprint- and iris-based authentication systems, require specialized and expensive equipment for acquiring biometric information. Furthermore, the system demands the active cooperation of a user. To alleviate the aforementioned drawbacks of fingerprint- and iris-based access control, human face recognition (HFR) has been increasingly applied to access control systems. The HFR-based access control has desirable characteristics: (i) it does not require specialized optical equipment and (ii) it provides user-friendly identification and authentication due to the non-intrusive and non-aggressive nature of HFR [1,2]. Salvador et al. [3] proposed an automatic reception desk demonstration system with regularized linear discriminant analysis (R-LDA)-based HFR for securing access control to facilities.

An HFR-based access control system using Fisher linear discriminating analysis (FDA) and principal component analysis (PCA) was introduced in Xueguang and Xiaowei [4]. Hadid et al. [5] developed an access control system based on face detection and face recognition with local binary pattern (LBP) features. The multi-modal access control systems with RFID and HFR were proposed to provide more strong secure access control [6,7].

Despite the desirable characteristics of HFR, a number of HFR-based access control systems do not fully provide user-friendly authentication due to limited computational resources and deteriorating environmental conditions. For example, some systems eliminate the face detection step by compelling users to align their face with the fixed region to verify a user identity. Consequently, the active cooperation of users is mandatory for user authentication. User-friendly face recognition has been studied for developing biometric systems in uncontrolled environments [8]. In general, a user-friendly HFR consists of a face detection step for facial image alignment, and face recognition for user identification and authentication.

To realize a user-friendly HFR, it should be able to detect faces and eyes of varied sizes ranging from small faces at a long distance to large faces at a short distance. At the Beijing 2008 Olympic Games, access control systems based on user-friendly HFR [8] were used to verify the identities of ticket holders. The system used AdaBoost and a multi-block local binary pattern (MB-LBP) for face detection and extracted MB-LBP and Gabor features for facial image description. The self-quotient image (SQI) technique was used to manage the illumination change problem.

HFR performance may deteriorate with changes in illumination, pose, and facial expression. Since access control systems are installed in various environments that are both indoors and outdoors, this inevitably involve changes in illumination, where effect of illumination constitutes the most challenging issue for HFR-based access control systems. To manage the aforementioned problem, there are on-going research efforts in the field of 3D face recognition and analysis. 3D face recognition methods are able to accurately recognize the face and detect facial landmarks under an illumination change with varied facial positions and expressions. For the acquisition of 3D facial data, specialized hardware equipment, such as a 3D scanner and 3D camera, is required [9,10,11,12]. One of the most challenging issues in the face recognition domain is to find an efficient and discriminative facial image descriptor. The most widely used methods are based on subspace analysis and statistical learning, i.e., PCA and FDA. These approaches are sensitive to varying illuminations, pose, and expression changes [13,14].

To manage difficulties in subspace analysis and statistical learning methods, non-statistical learning methods, e.g., Gabor wavelet and LBP, have been employed for facial image descriptions. The Gabor-wavelet-based facial features are robust to noise and local image transformations caused by changes in lighting, occlusions, and head poses [15,16]. An LBP histogram is computationally efficient, nonparametric, and invariant to monotonic changes in illumination [17]. Combinational models consisting of a Gabor wavelet and an LBP histogram [18,19], namely Gabor-LBP histograms, have received a considerable amount of attention. With the great success of Gabor wavelets and LBP histograms in HFR, there are many variations of Gabor-wavelet- and LBP-based face representation [16,20,21]. Despite these facial representations showing significantly good performance, the dimensionality of the feature vector is too high.

As a summary, a user-friendly access control system based on HFR should satisfy the following requirements: (i) it should be able to detect faces and eyes of varied sizes at a distance, (ii) it should be robust and invariant to changes in illumination, and (iii) the facial image descriptor should be lightweight and discriminative.

The goal of this study is to provide a user-friendly face recognition approach for an access control system that meets all the above system requirements for user-friendly access control. This study employed our previous work on face detection algorithms based on the LBP and AdaBoost [22], which are fast and robust to changes in illumination, and modified the framework to additionally detect and localize eyes. Using multi-level pyramid images, the system could detect faces and eyes of varied sizes at a distance. The proposed system uses a lightweight and effective face representation model combining Gaussian derivative filters with the LBP histogram [20,22], which is invariant to changes in illumination and complicated backgrounds. For fast and accurate face matching, the histogram intersection is used as a score function. To validate the performance of the proposed system, we conducted experiments using a benchmark dataset and video clips collected in an indoor environment. The experimental results show that the proposed system achieved feasible accuracy for practical applications and was straightforward to implement with a reasonable execution time.

The remaining parts of this paper are organized as follows. In Section 2, we provide the related works. The proposed user-friendly face recognition approach for access control is provided in Section 3. In Section 4, experimental results and discussion are provided. Finally, some concluding remarks are given in Section 5.

## 2. Related Work

With the growing popularity of smart/embedded devices in consumer electronics, there has been an increasing trend of using HFR to identify and to verify a person for consumer applications, such as smart home appliances [23], security of a mobile device [24], and access/border control. Zuo and de With [23] proposed a near-real-time face recognition system for embedding in smart home applications. Kremic, Subasi, and Hajdarevic [24] implemented a client-server mobile application for face recognition for identity authentication for access control and prevention of unauthorized mobile device usages. An automatic reception desk demonstration system based on HFR was presented in Salvador and Foresti [3]. An HFR-based access control system using Fisher linear discriminating analysis (FDA) and principal component analysis (PCA) was introduced in Xueguang and Xiaowei [4]. Hadid, Heikkila, Ahonen, and Pietikainen [5] developed an access control system based on face detection and face recognition with the LBP features. The multi-modal access control systems with RFID and HFR were proposed to provide stronger secure access control. For the Beijing 2008 Olympic Games, the access control system based on HFR at a distance was used to verify the identities of the ticket holders [8].

Saraf et al. [25] proposed an approach for an automatic stand-alone door access control system based on face recognition, which was developed by using a Raspberry Pi electronic development board and was operated on a battery power supply and wireless internet connectivity by using a USB modem. Boka et al. [26] developed a hardware and software system for access monitoring, which was implemented on a Raspberry Pi with two portable cameras using the Intel Movidius Neural Compute System. They used a deep learning model, namely FaceNet. Baksshi et al. [27] proposed the real-life application of a security lock system using a principal-component-based face recognition approach. They implemented the proposed system on an Arduino Uno microcontroller using an integrated webcam and MATLAB for the GUI. Sagar et al. [28] presented a smart locking system that chooses the algorithm for face detection and recognition based on the intensity of light at that time. They employed basic principal component analysis, linear discriminant analysis, and its variations for face detection and recognition. Sajjad et al. [29] proposed a Raspberry-Pi- and cloud-assisted face recognition framework, which employed bag of words for the extraction of oriented Features from Accelerated Segment Test (FAST) and rotated Binary Robust Independent Element Features (BRIEF) points [30] from the detected face, and a support vector machine for the identification of suspects.

The core steps of HFR are the face/eyes detection [31,32] and face representation [15,16,17,18,19,20,21]. Face and eyes detection are significant elements of HFR at a distance and are used for facial image alignment. The boosted cascade structure [21] proposed by Viola and Jones is most widely used for face and eyes detection. Gabor-wavelet- and LBP-based face representations have achieved great success in face recognition applications. Gabor-wavelet-based facial features are robust to noise and local image transformations due to lighting changes, occlusions, and pose variations [20]. The LBP histogram is computationally efficient, nonparametric, and invariant to monotonic illumination changes [17,33,34]. The combinational models including Gabor wavelets and LBP histograms [18,19], namely Gabor-LBP histograms, have received significant attention. With the great success of Gabor wavelets and LBP histograms in HFR, there are many variations of Gabor-wavelet- and LBP-based face representation [20,21].

A deep learning approach, a convolutional neural network (CNN) in particular, has been widely applied to facial image recognition applications. Zhang et al. [35] proposed a deep cascaded multitask framework for face detection and alignment in an unconstrained environment. The proposed approach exploits the inherent correlation between detection and alignment, and also leverages a cascaded architecture with three stages of deep convolutional networks to predict face and landmark location in a coarse-to-fine manner. Schroff et al. [36] presented a unified system, called FaceNet, for face verification, recognition, and clustering, which is based on learning a Euclidean embedding per image using a deep convolutional neural network. With FaceNet embedding as a feature vector, face verification, recognition, and clustering can be easily solved using standard techniques, such as thresholding, k-NN classification, and k-means clustering. Chen et al. [37] proposed an efficient CNN model, called MobileFaceNets, for accurate real-time face verification on mobile devices. The proposed method uses no more than 1 million parameters.

On the other hand, 3D face recognition has been studied in various application domains, such as surveillance, homeland security, entertainment, and medical domains. With the fast evolution of 3D sensors, 3D face recognition could overcome the fundamental limitations of conventional 2D face recognition. The geometric information of a 3D facial dataset could not only improve the recognition accuracy under unconstrained conditions, which are difficult for 2D face recognition, but could also be used for medical analysis [9,10]. For example, Marcolin et al. [11] proposed a 3D landmarking algorithm that is able to extract eyebrows and mouth fiducial points in multi-expression face analysis. By extracting the eyebrows and mouth landmarks from various face poses, they provide the possibility of the robust and pose-independent facial analysis in the field of medical application. Moos et al. [12] provide a study on computer-aided morphological analysis for maxilla-facial diagnostics, which compares the 3D morphometric methods, such as conventional cephalometric analysis (CCA) and generalized Procrustes superimposition (GPS), with principal component analysis (PCA), thin-plate spline analysis (TPS), multisectional spline (MS), and clearance vector mapping (CVM), while applying them to a case study of five patients affected by the malocclusion pathology.

## 3. User-Friendly Face Recognition

### 3.1. Face Recognition Methods for Access Control

This subsection provides the lightweight and computationally efficient face recognition method for access control, which consists of a face detection and eye localization step for facial image alignment, a pre-processing step for facial image normalization and illumination normalization, a facial feature extraction step, and a face recognition and user identification step. The proposed approach is invariant to complicated illumination changes and backgrounds. The details of each step are provided in the following subsections. The overall procedure is illustrated in Figure 1.

### 3.2. Face Detection and Eye Localization

Facial image alignment is a significant step in HFR systems and consists of face detection and eye localization. The facial region of the input image is detected to reduce the region of interest (ROI) for eye detection. The precise eye positions are detected in the reduced ROI for eye localization. Using the localized eye positions, facial images are aligned to apply the face recognition algorithm. To provide user-friendly access control, the capability of face and eye detection at a distance with less user cooperation is required. It should be able to detect faces and eyes of varied sizes ranging from small faces at a long distance to large faces at a short distance.

The proposed system uses the LBP-AdaBoost approach [17,32,38] for face detection and eye localization. This method follows the boosted cascading structure of Viola and Jones [17,39], and adopts the LBP histogram as feature vectors, and the two-class LDA as weak classifiers, respectively. With multi-level pyramid images (i.e., resized input images), it is possible to detect faces and eyes of varied sizes at a distance. The local texture and shape of face images are presented as LBP features obtained the through the ${\mathrm{LBP}}_{8,R}^{u2}$ operator. We used 21 multi-level pyramid images from original input images with 256 by 256 pixels. The size of the pyramid image at a certain level can be obtained using:(1)$$\left(\mathrm{resised}\;\mathrm{width},\;\mathrm{resized}\;\mathrm{height}\right)=\;\frac{\left(\mathrm{width},\;\mathrm{height}\right)}{\exp},$$
where $\exp=\;{\mathrm{scale}}^{\left(\mathrm{pyr}\_\mathrm{level}-1\right)}$.

In this paper, we set the scale to 1.1 and varied the pyr_level from 10 to 30. The proposed approach could detect faces of not less than 32 by 32 pixels and of not less than 16 by 16 pixels in each pyramid image. We used LDA for discriminative weak classifier, which represents the local image structure of the sub-region within a face image [32,38,39]. We designed 30 weak classifiers in a similar fashion to Viola’s approach [32,39].

By employing the LBP features, it is invariant to monotonic changes of illumination and is computationally efficient. The overall architectures of LBP-AdaBoost for face and eye detection are illustrated in Figure 2. Figure 3 presents the examples of positive and negative training sets for face and eyes detection.

### 3.3. Face Recognition Based on a Gaussian Derivative Filter-LBP Histogram

Once the face and eyes have been detected, the facial region of the input image is normalized to a fixed size based on the localized eye positions. For a rotated face, the image is rotated such that the eyes are horizontal. The distance between the eyes is normalized to 32 pixels and the face is cropped to a 64-by-64-pixel size. Although LBP-histogram-based face representation has desirable characteristics, such as computational efficiency, non-parametric operation, and invariance to monotonic changes of illumination, it is sensitive to random noise and non-monotonic changes of illumination [16]. To overcome the effects of complicated illumination changes and backgrounds, a self-quotient image (SQI) [40] is used for the pre-processing of normalized facial images. SQI is defined as the ratio of the input image and its smooth versions, which is derived from one image and has the same quotient form as that in the quotient image method.

The Gabor-LBP histogram framework [18,19] was modified to enhance the representation power of the spatial histogram. Forty Gabor wavelet filters were replaced with eight first-order and eight second-order Gaussian derivative filters in the face representation framework since their parameters are easier to manipulate, and better recognition performances have been shown with a smaller number of filters [20]. The first-order and second-order Gaussian derivative filters for a facial image description are derived as follows [20,22]: a two-dimensional Gaussian function G(x,y) is given as:(2)$$G(x,y)=\exp\left(-\frac{x^{2}}{2\sigma_{x}^{2}}-\frac{y^{2}}{2\sigma_{y}^{2}}\right),$$
where $\sigma_{x}$ and $\sigma_{y}$ are the standard deviation parameters.

From Equation (2), the first-order (Equation (3)) and second-order (Equation (4)) Gaussian derivative filters in the θ–orientation can be obtained using:(3)$$F(x,y,\theta )=-\frac{x_{\theta }}{\sigma_{x}^{2}}\exp\left(-\frac{x_{\theta }^{2}}{2\sigma_{x}^{2}}-\frac{y_{\theta }^{2}}{2\sigma_{y}^{2}}\right),$$
(4)$$S(x,y,\theta )=\left(\frac{x_{\theta }^{2}-\sigma_{x}^{2}}{\sigma_{x}^{4}}\right)\exp\left(-\frac{x_{\theta }^{2}}{2\sigma_{x}^{2}}-\frac{y_{\theta }^{2}}{2\sigma_{y}^{2}}\right),$$
where $x_{\theta }=x\cos\theta -y\sin\theta$ and $y_{\theta }=x\sin\theta -y\cos\theta$.

Using Gaussian derivative filters, the dimensions of the facial features can be reduced by 60% compared with the Gabor wavelet filters. The derived first-order and second-order Gaussian derivative filters are illustrated in Figure 4.

A facial image is presented as a histogram sequence using the following steps. First, the multiple Gaussian derivative filter images are transformed by convolving the normalized facial images with the first-order and second-order Gaussian derivative filters. Let I(x,y) be the input facial image; its convolution with the Gaussian derivative filters $GF_{d,\theta }\left(x,y\right)$ can be defined as:(5)$$Conv_{I,GF}(x,y,d,\theta )=I(x,y)*GF_{d,\theta }(x,y),$$
where *d* and θ are the order of the derivative and orientation, respectively.

The $GF_{d,\theta }\left(x,y\right)$ can be either Equation (3) or Equation (4) with respect to the order of the derivatives. The symbol ∗ denotes the convolution operator. Figure 5 shows examples of multiple Gaussian derivative filter images.

Second, each Gaussian derivative filter image is converted into an LBP map image using the LBP operator. The LBP operator [17] labels the image pixels by thresholding the 3 × 3-neighborhood of each pixel $P_{i}$ (*i* = 0, 1, 2, …, 7) with the center value $P_{c}$ and by considering the result as a binary number:(6)$$T\left(p_{i}-p_{j}\right)=\{\begin{array}{c} 1,\;p_{i}\geq p_{c}\\ 0,\;p_{i}<p_{c} \end{array}$$

The LBP patterns at each pixel can then be achieved by summing the threshold values weighted by a power of two, which characterizes the spatial structure of the local image texture:(7)$$LBP=\sum_{i=0}^{7}T(p_{i}-p_{c})\cdot 2^{i}$$

Figure 6 shows examples of the LBP map images obtained by applying the LBP operator to the multiple Gaussian derivative filter images shown in Figure 5.

Finally, each LBP map image is divided into non-overlapping sub-regions with a predefined bin size, and the histograms of the sub-regions are computed. A total of 36 subregions for each LBP map image are used. The LBP histograms of the sub-regions are concatenated to form the final histogram sequence, which serves as the face representation.

For face matching, the histogram intersection [16,20,41] is used as a score matching function of two facial features since it is simple, fast, and shows good recognition performance. The final decision for user identification is made by thresholding the matching score:(8)$$FM\left(H_{A},H_{B}\right)=\sum_{i=1}^{subregions}\min\left(H_{A}\left(i\right),H_{B}\left(i\right)\right),$$
where $H_{A}$ and $H_{B}$ denote the facial features, i.e., the sequence of histogram, and *i* represents the *i*th subregion.

### 3.4. User-Friendly Access Control System

This subsection presents the overall architecture of the proposed user-friendly access control system based on dual-factor authentication i.e., RFID token and HFR, which verifies the identity of the RFID card holder to prohibit an unauthorized user from gaining access by obtaining an authorized RFID card. The system mainly consists of four parts: (i) face detection at a distance, (ii) facial feature extraction, (iii) user ID recognition, and (iv) user authentication as shown in Figure 7.

The system provides two different operation modes, i.e., user registration mode and user authentication mode. In user registration mode, the facial images of a user are acquired by a camera, and then the face and eyes detector finds the facial region and localizes the eyes’ positions. The detected face images are normalized with localized eyes positions. From a normalized face image, the facial feature is extracted by means of Gaussian derivative filters and a LBP histogram. While the facial feature is extracted, the system reads the user ID from the RFID card that is held by the user. The extracted facial features and user ID are stored in the database (DB) for user authentication.

In user authentication mode, the system extracts facial features (namely, probe facial features) from acquired facial images of a user in precisely the same way as the user registration mode while a user is approaching the system. When a user is close to the system, i.e., from three meters to one meter, the system reads the user ID from the RFID card and validates it. If the user ID is in the registration lists of the system, the corresponding facial features (namely, gallery facial features) stored in the DB are loaded for user authentication. Otherwise, the user access is denied. The system is able to authenticate a user while he or she is walking. The authentication range is from three meters to one meter according to the user’s walking speed. Finally, the system matches the gallery facial features and the probe facial features, and computes the matching score as a similarity measurement. The final decision for granting the access right is made by thresholding the matching score.

The hardware configuration of the proposed stand-alone access control system is provided in Figure 8. It consists of an optical camera, remote controller, RFID reader, proximity sensor, control button interface, audio codec, speaker, and memory devices. This study used an optical camera with a USB interface as an image acquisition device to get high quality face images. It also used an active-type RFID reader with a proximity sensor to provide a user-friendly interface. A user does not need to touch the RFID reader to gain access rights because the system reads the user ID from the active RFID card at a distance when the proximity sensor detects a user. The remote controller emits the signal to open the gate when an authorized user is close to the system. The distance between a user and the system is computed via calibration of the proximity sensing values. The system alerts significant events, such as user registration completion, user denial, and user authorization using audio voices.

## 4. Experimental Results

### 4.1. Experimental Setup

The proposed user-friendly, stand-alone access control system based on HFR at a distance with RFID was implemented using the embedded evaluation board with a dual core CPU, 2.50 DMIPS/MHz per core. The mockup of the proposed system, viz. BioGate access control system, is provided in Figure 9. The system used a camera with a USB interface as an image acquisition device to obtain high-quality facial images. It also uses an active RFID reader equipped with a proximity sensor to provide a user-friendly interface. The remote controller transmits the signal to open the gate once the user has been authorized.

We employed the true acceptance rate (TAR), the false rejection rate (FRR), the false acceptance rate (FAR), and the equal error rate (EER) [38,42,43,44,45,46] as performance evaluation measurements. The TAR is defined as the frequency at which the system correctly verifies a true claim of authorization. The FRR is the frequency of the system rejecting a true claim of authorization. The FAR is defined as the frequency of the system incorrectly accepting an unauthorized user [38,46,47,48,49,50,51,52]. The ERR is defined as the error rate when FAR and FRR are equal for a particular threshold [44]. FNMR(*t*) and FMR(*t*) are defined as the distribution of the false non-match rate (FNMR) and false match rate (FMR) with threshold value *t* varying from 0.0 to 1.0. FNMR is the fraction of genuine comparisons (a user is compared with gallery images of her/himself) that produce a matching score of less than threshold value *t*. FMR is the fraction of impostor comparisons (a user is compared against gallery images of all other users) that produce a matching score greater than or equal to threshold value *t* [45,46].

To evaluate the general face recognition performance of the proposed approach, experiments were conducted on the E-face [16] and XM2VTS [47] datasets. The E-face dataset consisted of 55 subjects with 20 images of each subject. The facial images were taken under different conditions, i.e., different lighting conditions, variations in pose and facial expression, and with or without spectacles. Ten images per subject were used as gallery images, whereas the rest of the images were used as probe images. The XM2VTS dataset is one of the most widely used benchmark datasets and consists of 295 subjects with eight images per subject. Four images per subject were used as gallery images and the rest were used as probe images. In this experiment, the k-nearest neighborhood classifier with the histogram intersection similarity metric was employed.

To validate the proposed method’s robustness to changes in illumination, the experiment was conducted on the PF07 [48] database. The PF07 dataset includes 100 male and 100 female subjects with 320 images per each subject. The images were captured for five different poses under sixteen types of illumination with four different facial expressions. This experiment used facial images with a frontal facial pose, neutral facial expression, and sixteen illumination variations. One image (with no lighting) per subject was used as a gallery image, and fifteen images (with fifteen different lighting conditions) per subject were used as probe images. Examples of the E-face, XM2VTS, and PF07 face datasets are given in Figure 10.

To evaluate the overall performance of the proposed system, facial images for registration (as a gallery dataset) and videos for testing (as a probe dataset) were collected in an indoor environment, as illustrated in Figure 11. The newly collected dataset consisted of ten subjects with ten still images and one video sequence of each subject. The still images were captured at a two-meter distance from the system camera, and the videos were recorded while the subject was walking toward the system from a five-meter distance. The videos were recorded a few days after the still images were captured. Ten still images per subject were used as the gallery dataset, and twenty image frames per video were used as the probe dataset.

### 4.2. Experimental Evaluations

To show the face recognition performance of the proposed method, we compared our method with the Gabor-LBP histogram-based approach. Table 1 shows that the proposed face recognition approach achieved a better recognition accuracy with smaller facial feature vectors compared with the Gabor-LBP histogram-based approach. As shown in Figure 10a, the E-face dataset contains complicated backgrounds and variations of facial image conditions, such as illumination changes, small variations of pose and facial expression, and the presence or absence of spectacles. This experiment indicated that the proposed approach was more invariant to complicated backgrounds and variations of facial image conditions than the Gabor-LBP histogram-based approach.

The cumulative accuracies on E-face and XM2VTS face databases are shown in Figure 12. The experimental results showed that the proposed method was suitable for not only face identification, but also face retrieval application.

Table 2 provides a performance evaluation of the proposed approach on the PF07 database. The system achieved a TAR of 99.53%, an FRR of 0.47%, and a FAR of 0.08%. The threshold value *t* was set to 0.64. Figure 13 illustrates the results of the performance evaluation on the PF07 database.

The distribution of the false non-match rate (FNMR) and false match rate (FMR) with threshold value *t* varying from 0.0 to 1.0 is presented in Figure 13a. The proposed method achieved an EER of 0.28%, as shown in Table 2 and Figure 13a. The receiving operation curve (ROC) is shown in Figure 13b. The FNMR is plotted as a function of FMR, and the curve is drawn at a log-log scale for better comprehension.

Table 3 shows the results of an evaluation of the overall performance of the proposed system on newly collected facial images (for a gallery dataset) and videos (for a probe dataset) under an indoor office environment, as shown in Figure 11. The system achieved a TAR of 95.0%, an FRR of 5.0%, an FAR of 0.0%, and an EER of 2.33% at the acceptance threshold value *t* = 0.63.

The average execution time of the proposed system for processing one image frame is presented in Table 4. The input image was a gray image of 640 × 480 pixels. The total execution time included the I/O processing times of the peripheral devices. The proposed system was able to process at 5–6 fps.

We summarize the quantitative and qualitative analysis with existing methodologies in Table 5. The deep-learning-based approach [26] shows superior performance in terms of accuracy over the other approaches. To apply a deep-learning-based approach to an embedded system, a specialized neural computing device, such as the Intel Movidius Neural Compute Stick, is needed. Many systems are implemented using Raspberry Pi and employ handcrafted features.

## 5. Conclusions

This paper presented the design and implementation of a multimodal, stand-alone access control system based on human face recognition at a distance with RFID technology to provide a user-friendly interface and more secure access control. First, the implementation issues of stand-alone access control systems based on human face recognition were introduced as follows: (1) lack of computational resources, (2) the requirement of active user cooperation, and (3) sensitivity to illumination changes. To manage these implementation issues, a fast and lightweight face recognition approach was proposed, which consists of a face and eyes detection step and a face recognition step. For the face and eyes detection, this paper adopted the LBP-AdaBoost approach that is fast and robust to illumination changes.

The proposed method could detect the face and eyes of various image sizes at a distance in a way that made the system more user-friendly. The system combined Gaussian derivative filters with LBP histograms for face representation, which is lightweight and invariant to complicated illumination changes and background compared with the Gabor-LBP-based facial features. The proposed system achieved a true acceptance rate of 95.0%, with a false acceptance rate of just 0.0%, and an average processing speed of 5.26 fps regarding newly collected facial images and video datasets in an indoor environment.

## Figures and Tables

**Figure 1 sensors-20-00785-f001:**
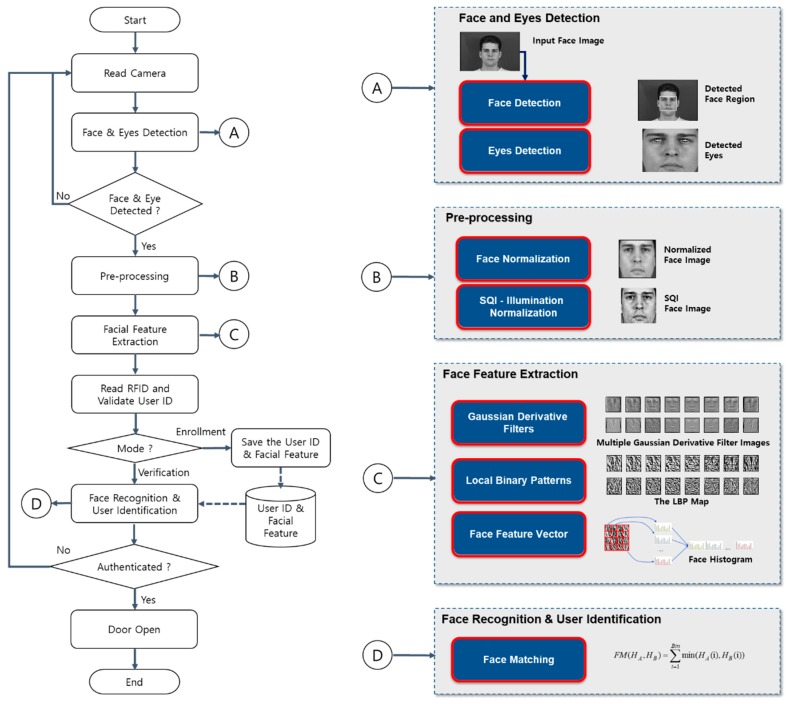
Overall process of the face recognition for a stand-alone access control system. RFID: ???, SQI: self-quotient image.

**Figure 2 sensors-20-00785-f002:**
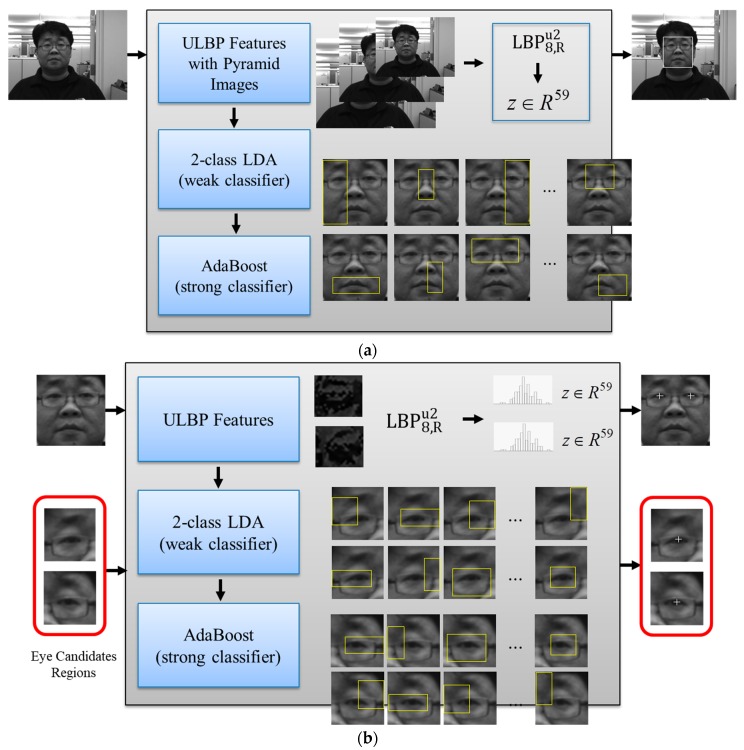
The local binary pattern (LBP)-AdaBoost approaches to face detection (**a**) and eye detection (**b**) for facial image alignment. LDA: linear discriminant analysis.

**Figure 3 sensors-20-00785-f003:**
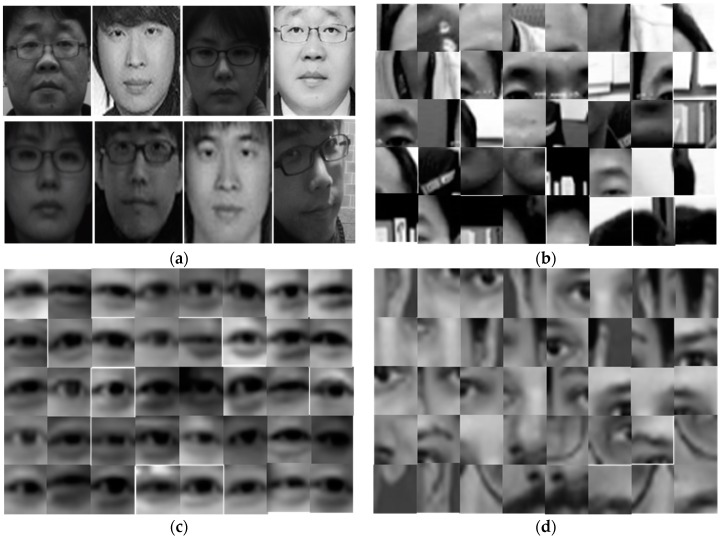
Examples of a training dataset for face and eye detection: (**a**) positive training set for face detection, (**b**) negative training set for face detection, (**c**) positive training set for eye detection, and (**d**) negative training set for eye detection.

**Figure 4 sensors-20-00785-f004:**
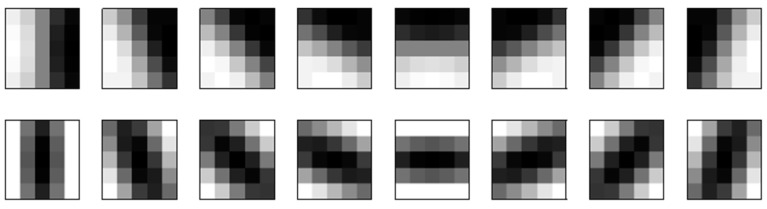
First-order and second-order Gaussian derivative filters for facial feature descriptions. The figures in the first row are first-order Gaussian derivative filters, and those in the second row are second-order Gaussian derivative filters.

**Figure 5 sensors-20-00785-f005:**
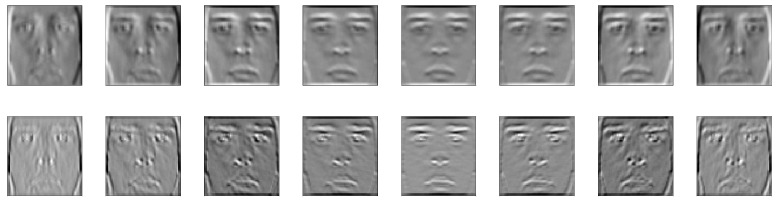
Examples of multiple Gaussian derivative filter images obtained by convolving normalized facial images with the Gaussian derivative filters shown in Figure 4.

**Figure 6 sensors-20-00785-f006:**
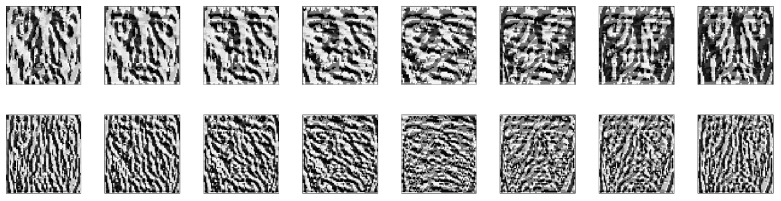
Examples of the LBP map images obtained by applying the LBP operator to the multiple Gaussian derivative filter images shown in Figure 5.

**Figure 7 sensors-20-00785-f007:**
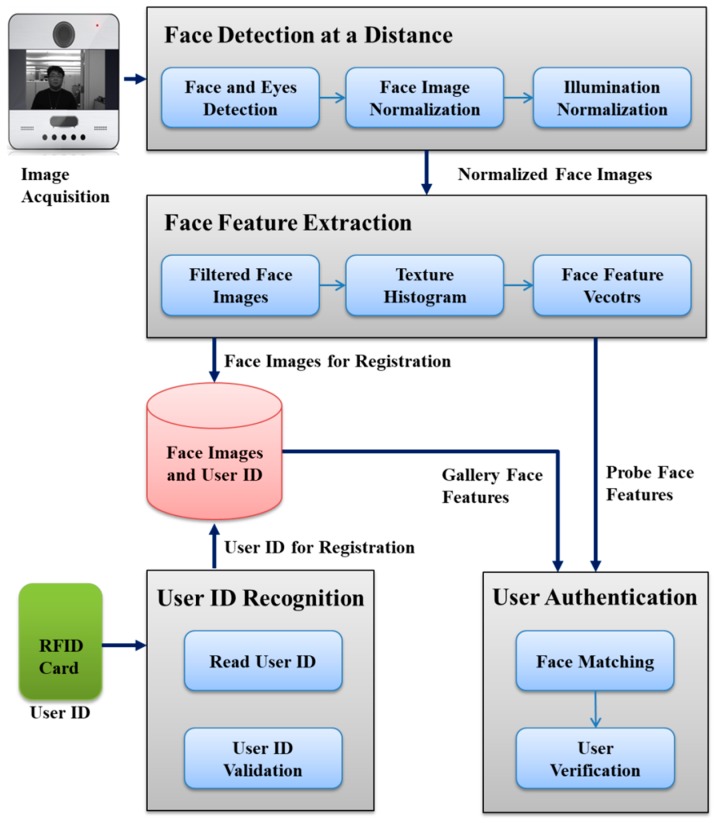
Overall architecture of the proposed user-friendly, stand-alone access control system based on human face recognition at a distance with RFID.

**Figure 8 sensors-20-00785-f008:**
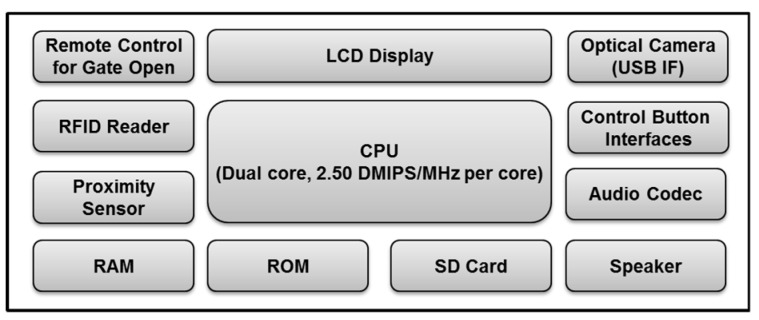
The hardware configuration of the proposed user-friendly stand-alone access control system based on human face recognition at a distance with RFID. DMIPS, IF: interface.

**Figure 9 sensors-20-00785-f009:**
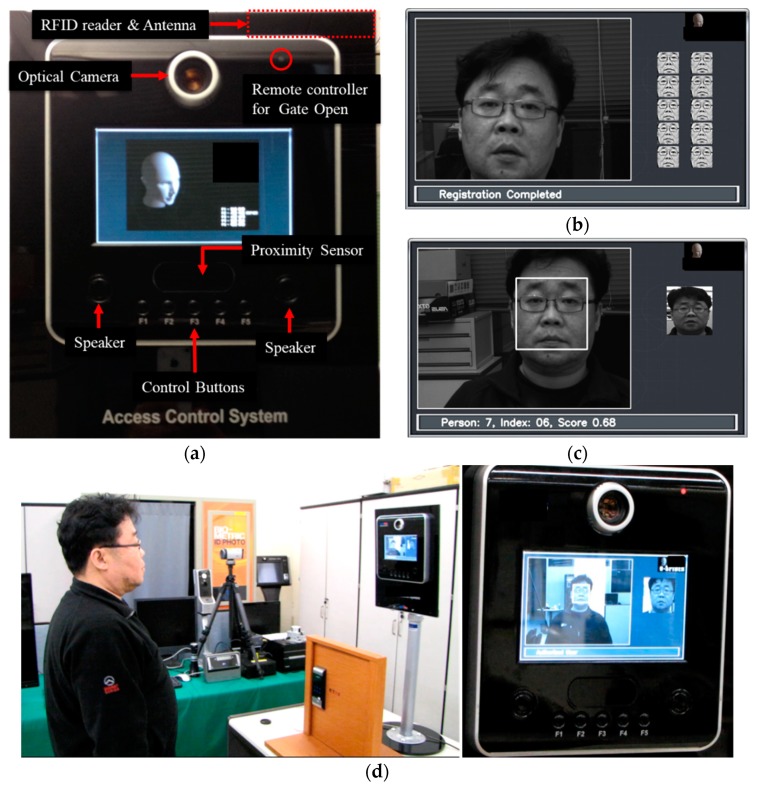
The mockup of the proposed user-friendly, stand-alone access control system based on HFR at a distance with RFID, viz. the BioGate access control system: (**a**) system configuration of the mockup system, (**b**) example of user registration, (**c**) example of user authentication, and (**d**) example of the experimental environment.

**Figure 10 sensors-20-00785-f010:**
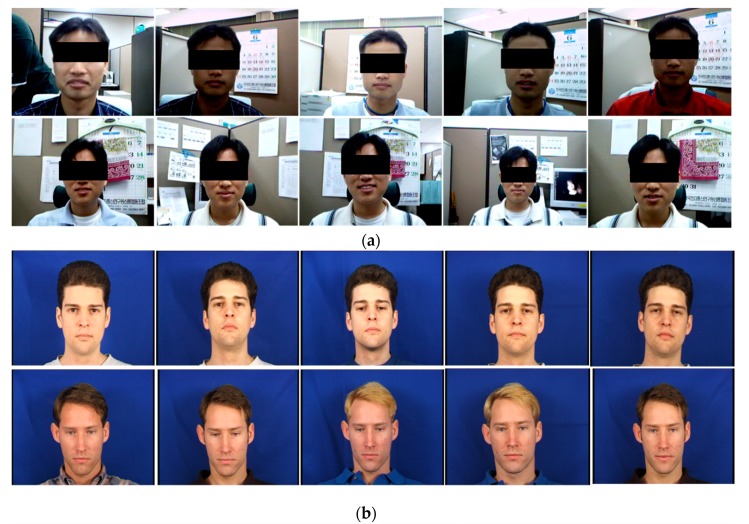
Examples of face databases for experiments: (**a**) examples in the E-face database; (**b**) examples in the XM2VTS face database; and (**c**) examples in the PF07 database with frontal face pose, neutral face expression, and 16 illumination variations.

**Figure 11 sensors-20-00785-f011:**
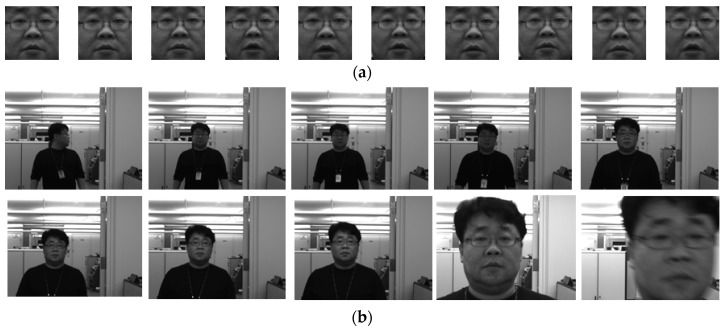
Examples of newly collected facial images (for a gallery dataset) and videos (for a probe dataset) in an indoor office environment: (**a**) examples of registered images normalized from gallery images and (**b**) examples of probe images extracted from the videos.

**Figure 12 sensors-20-00785-f012:**
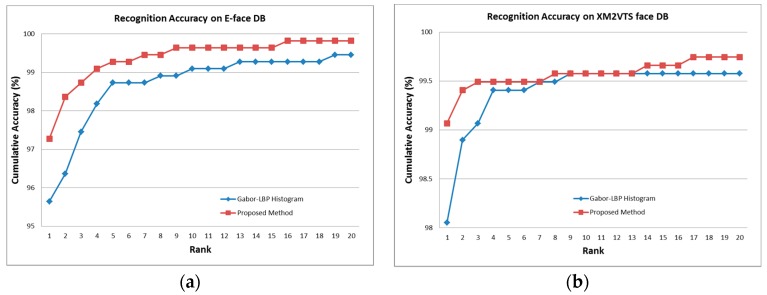
Cumulative accuracies of face recognition on the E-face database (**a**) and XM2VTS face database (**b**).

**Figure 13 sensors-20-00785-f013:**
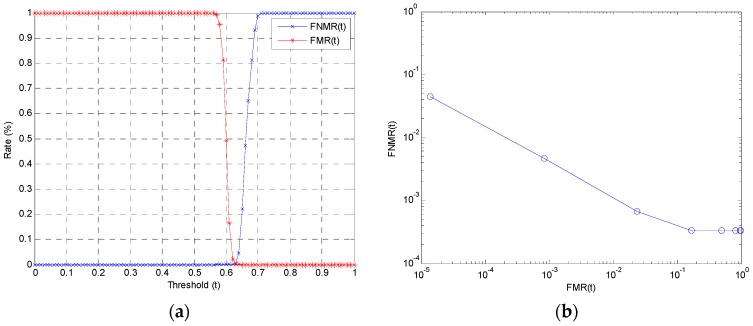
Illustration of the performance evaluation on the PF07 database: (**a**) the distribution of false not-matching rate (FNMR) and false matching rate (FMR) with threshold value *t* varying from 0.0 to 1.0, and (**b**) the log-log scale receiving operation curve (ROC) with the false not-matching rate and false matching rate.

**Table 1 sensors-20-00785-t001:** Recognition rate (%) of the proposed face recognition approach compared with the Gabor-LBP histogram-based approach on the E-face and XM2VTS face databases.

Face Database (DB)	Gabor-LBP Histogram	Proposed Method
**E-face Database**	**95.64%**	97.27%
XM2VTS Face Database	98.05%	99.06%

**Table 2 sensors-20-00785-t002:** Performance evaluation of the proposed face recognition approach on the sampled PF07 database containing sixteen different illuminations using the following criteria: TAR—true acceptance rate, FRR—false rejection rate, FAR—false acceptance rate, EER—equal error rate.

Performance Criterion	TAR	FRR	FAR	EER
Performance	99.53%	0.47%	0.08%	0.28%

**Table 3 sensors-20-00785-t003:** Performance evaluation of the proposed access control system on newly collected facial images and videos in an indoor office environment.

Performance Criterion	TAR	FRR	FAR	EER
Performance	95.00%	5.0%	0.00%	2.33%

**Table 4 sensors-20-00785-t004:** The average execution times (millisecond) of the proposed access control system for processing one image frame.

Process	Average Execution Time
Face and Eyes Detection	102 ms
Face Verification	53 ms
Total Execution Time	190 ms

**Table 5 sensors-20-00785-t005:** Comparison with the stand-alone access control systems based on facial image recognition.

Criteria	Saraf et al. [25]	Boka et al. [26]	Bakshsi et al. [27]	Sagar et al. [28]	Sajjad et al. [29]	Proposed Method
Embedded H/W	Raspberry Pi	Raspberry Pi with the Intel Movidius Neural Compute Stick	Arduino Uno	Renesas board (RL78 G13)	Raspberry Pi 3 model B, ARM Cortex-A53, 1.2 GHz processor, video Core IV GPU	Cortex dual core CPU, 2.50 DMIPS/MHz per core
Camera	Single camera	Two cameras	Single camera	Single camera	Raspberry Pi camera	Single camera
Face detection	Haar cascade method	Haar-feature based approach	N/A	PCA, LDA, histogram equalization	Viola Jones Method	AdaBoost with LBP
Facial feature	LBP histogram	FaceNet	Global feature with principal component analysis	PCA, LDA, histogram equalization	Bag of words with oriented FAST and rotated BRIEF	Gaussian derivative filter-LBP histogram
Face identification	N/A	k-NN	Euclidean Distance	N/A	Support vector machine	Histogram intersection
Execution time	N/A	N/A	N/A	10 sec for authentication	9 s for first 10 matches	190 ms per image
Overall accuracy	N/A	99.60% accuracy on LFW dataset	N/A	90.00% accuracy on own dataset	91.33% average accuracy on Face-95 dataset	95.00% true accept rate on own video stream dataset

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
