# Peer review of "Face Recognition at a Distance for a Stand-Alone Access Control System†"

_sensors, 2020, doi:10.3390/s20030785_

Round 1

Reviewer 1 Report

This article presents a face recognition system based on handcrafted features. This work completely ignores the recent years of development in this research field and focuses mostly on obsolete references. Many publicly available works based on Convolutional Neural Networks would certainly outperform this work, including the most common baseline nowadays (MTCNN [1] + FaceNet [2]). Besides, this work lacks a comparison to the state-of-the-art, which makes it unacceptable in the current form.

[1] Kaipeng Zhang, Zhanpeng Zhang, Zhifeng Li, Yu Qiao: Joint Face Detection and Alignment using Multi-task Cascaded Convolutional Networks.

[2] Florian Schroff, Dmitry Kalenichenko, James Philbin: FaceNet: A Unified Embedding for Face Recognition and Clustering.

Author Response

Comments and Suggestions for Authors

This article presents a face recognition system based on handcrafted features. This work completely ignores the recent years of development in this research field and focuses mostly on obsolete references. Many publicly available works based on Convolutional Neural Networks would certainly outperform this work, including the most common baseline nowadays (MTCNN [1] + FaceNet [2]).

[1] Kaipeng Zhang, Zhanpeng Zhang, Zhifeng Li, Yu Qiao: Joint Face Detection and Alignment using Multi-task Cascaded Convolutional Networks.

[2] Florian Schroff, Dmitry Kalenichenko, James Philbin: FaceNet: A Unified Embedding for Face Recognition and Clustering.

â–º Response: Thank you for your kind comment. Yes, we added recent research works to the references, and cited the paper in Related Works section as follows:

Deep learning approach, in particular Convolutional Neural Network (CNN), has been widely applied to the facial image recognition applications. Zhang et al. [35] propose a deep cascaded multitask framework for face detection and alignment under unconstrained environment. The proposed approach exploits the inherent correlation between detection and alignment, and also leverages a cascaded architecture with three stages of deep convolutional networks to predict face and landmark location in a coarse-to-fine manner. Schroff et al. [36] present a unified system, called FaceNet, for face verification, recognition and clustering, which is based on learning a Euclidean embedding per image using deep convolutional neural network. With FaceNet embedding as feature vector, face verification, recognition and clustering can be easily solved using standard techniques such as thresholding, k-NN classification, and k-means. Chen et al. [37] propose an efficient CNN models, called MobileFaceNets, for accurate real-time face verification on mobile devices. The proposed method use no more than 1 million parameters.

K. Zhang, Z. Zhang, Z. Li, Y. Qiao, “Joint Face Detection and Alignment using Multi-task Cascaded Convolutional Networks,” IEE Signal Processing Letters, vol. 23, issue 23, pp. 1499-1503, 2016. F. Schroff, D. Kalenichenko, and J. Philbin, “FaceNet: A unified embedding for face recognition and clustering,” in Proc. CVPR, Boston, MA, USA, June 2015. S. Chen, Y. Liu, X. Gao, and Z. Han. (2018) MobileFaceNets: Efficient CNNs for Accurate Real-Time Face Verification on Mobile Devices. In: Zhou J. et al. (eds) Biometric Recognition. CCBR 2018. Lecture Notes in Computer Science, vol 10996. Springer, Cham

Besides, this work lacks a comparison to the state-of-the-art, which makes it unacceptable in the current form.

â–º Response: Thank you for your kind comment. We compared the proposed methods with existing state of art stand-alone access control system with all possible efforts. We added the summarization of the quantitative and qualitative analysis with other stand-alone access control system in the Experimental Results Section as follows:

We summarize the quantitative and qualitative analysis with existing methodologies in Table 5. The deep learning based approach [26] shows superior performance in accuracy over the other approaches. To apply deep learning based approach to the embedded system, specialized neural computing device such as the Intel Movidius Neural Compute Stick is needed. Many systems are implemented using Raspberry Pi and employ handcrafted features.

Table 5. Comparison to the standalone access control systems based on facial image recognition.

Criteria

Saraf et al. [25]

Boka et al. [26]

Bakshsi et al. [27]

Sagar et al. [28]

Sajjad et al. [29]

Proposed Method

Embedded H/W

Raspberry Pi

Raspberry Pi with the Intel Movidius Neural Compute Stick

Arduino UNO

Renesas board (RL78 G13)

Raspberry Pi 3 model B, ARM Cortex-A53, 1.2 GHz processor, Video Core IV GPU

Cortex dual core CPU, 2.50 DMIPS/MHz per core

Camera

Single Camera

Two Camera

Single Camera

Single Camera

Raspberry Pi Camera

Single Camera

Face Detection

Haar cascade method

Haar-feature based approach

N/A

PCA, LDA, Histogram Equalization

Viola Jones Method

AdaBoost with LBP

Facial Feature

LBP histogram

FaceNet

Global feature with PCA

PCA, LDA, Histogram Equalization

Bag of Words with Oriented FAST and Rotated BRIEF

Gaussian Derivative Filter-LBP histogram

Face Identification

N/A

k-NN

Euclidean Distance

N/A

SVM

Histogram Intersection

Execution Time

N/A

N/A

N/A

10 sec. for authentication

9 sec.

First 10 matches

190ms

Per image

Overall Accuracy

N/A

99.60% accuracy on LFW dataset

N/A

90.00% accuracy on own dataset

91.33% average accuracy on

Face-95 dataset

95.00% true accept rate on own video stream dataset

Reviewer 2 Report

The paper deals with an interesting topic face recognition for access control. In the present form the paper presents many lacks. Starting from the introduction section authors should provide a wider picture on the research domain, talking about not only 2D face analasys but also 3D reserches done, considering the last innovation on 3D camers introduced in many devices and the interesting relieability results obtained by these studies. Some more references should be added as for instance the following ones:

Marcolin, F. et al (2014). 3D Landmarking in multiexpression face analysis: a preliminary study on eyebrows and mouth. Aesthetic plastic surgery38(4), 796-811.

Moos, S. et al. (2010). Computer-aided morphological analysis for maxillo-facial diagnostic: a preliminary study. Journal of Plastic, Reconstructive & Aesthetic Surgery63(2), 218-226.

Regarding the methodological section authors should provide a more clear description of the methodology proposed by usage also of a graphical flowchart for going further with the details on the single methodology steps. Also the experimental validation should be improved, by the usage of a Design for Experiment (DOE) approach that could provide a more reliable outcome on the methodology performances

Author Response

Comments and Suggestions for Authors

The paper deals with an interesting topic face recognition for access control. In the present form the paper presents many lacks. Starting from the introduction section authors should provide a wider picture on the research domain, talking about not only 2D face analasys but also 3D reserches done, considering the last innovation on 3D camers introduced in many devices and the interesting relieability results obtained by these studies. Some more references should be added as for instance the following ones:

Marcolin, F. et al (2014). 3D Landmarking in multiexpression face analysis: a preliminary study on eyebrows and mouth. Aesthetic plastic surgery, 38(4), 796-811.

Moos, S. et al. (2010). Computer-aided morphological analysis for maxillo-facial diagnostic: a preliminary study. Journal of Plastic, Reconstructive & Aesthetic Surgery, 63(2), 218-226.

â–º Response: Thank you for your kind comment. Yes, we added recent research works to the references, and cited the paper in Introduction and Related Works section as follows:

In Introduction Section:

-------------------------------------------------- Omitted --------------------------------------------------------------

To deal with aforementioned problem, there are on-going research effort in the field of 3D face recognition and analysis. 3D face recognition methods are able to accurately recognize the face and detect facial landmarks under illumination change with variant facial positions and expression. For acquisition of 3D facial data, the specialized hardware equipment such as 3D scanner and 3D camera is required [9-12].

-------------------------------------------------- Omitted --------------------------------------------------------------

In Related Works Section:

-------------------------------------------------- Omitted --------------------------------------------------------------

On the other hand, 3D face recognition has been studied in various application domains such as surveillance, homeland security, entertainment and medical domains. With the fast evolution of 3D sensors, 3D face recognition could overcome the fundamental limitations of conventional 2D face recognition. The geometric information of 3D facial dataset could not only improve the recognition accuracy under unconstrained condition which are difficult for 2D face recognition, but also is used for medical analysis [9, 10]. For example, Marcolin et al. [11] propose a 3D landmarking algorithm which is able to extract eyebrows and mouth fiducial points in multi-expression face analysis. By extracting the eyebrows and mouth landmarks from various face pose, they provide the possibility of the robust and pose-independent facial analysis in the field of medical application. Moos et al. [12] provide a study on computer-aided morphological analysis for maxilla-facial diagnostic, which compares the 3D morphometric methods such as conventional cephalometric analysis (CCA), generalized Procrustes superimposition (GPS) with principal-components analysis (PCA), thin-plate spline analysis (TPS), multisectional spline (MS) and clearance vector mapping (CVM), while applying them to a case study of five patients affected by the malocclusion pathology.

-------------------------------------------------- Omitted --------------------------------------------------------------

S. Zhou and S. Xiao, “3D face recognition: a survey,” Human-centric Computing and Information Sciences, vol. 8, no. 35, 2018. S. Soltanpour, B. Boufama, and Q.M. Wu, “A survey of local feature methods for 3D face recognition,” Pattern Recognition, vol. 72, pp. 391-406, 2017. E. Vezzetti and F. Marcolin, “3D Landmarking in Multiexpression Face Analysis: A Preliminary Study on Eyebrows and Mouth,” Aesthetic plastic surgery, vol. 38, no. 4, pp. 796-811, 2014. E. Vezzetti, F. Calignano, and S. Moos, “Computer-aided morphological analysis for maxillo-facial diagnostic: a preliminary study,” Journal of Plastic, Reconstructive & Aesthetic Surgery, vol. 63, no. 2, pp. 218-226, 2010.

Regarding the methodological section authors should provide a more clear description of the methodology proposed by usage also of a graphical flowchart for going further with the details on the single methodology steps.

â–º Response: Thank you for your kind comment. We modified Figure 8 for clearing the overall description of the proposed algorithm as follows:

Figure 8. Overall process of the face recognition for a stand-alone access control system.

Also the experimental validation should be improved, by the usage of a Design for Experiment (DOE) approach that could provide a more reliable outcome on the methodology performances

â–º Response: Thank you for your kind comment. We split the Experimental Section into two subsection: 4.1 Experimental Setup and 4.2 Experimental Evaluation in order to provide the concise and clear explanation of experiment design and experimental evaluation. Also, we added the summarization of the quantitative and qualitative analysis with other stand-alone access control system in the Experimental Results Section as follows:

Experimental Results

4.1. Experimental Setup

The proposed user-friendly stand-alone access control system based on HFR at a distance with RFID was implemented using the embedded evaluation board with a dual core CPU, 2.50 DMIPS/MHz per core. The mockup of the proposed system, viz., BioGate access control system, is provided in Figure 9.

-------------------------------------------------- Omitted --------------------------------------------------------------

We employed the true acceptance rate (TAR), the false rejection rate (FRR), the false acceptance rate (FAR), and the equal error rate (EER) [38, 44-46] as performance evaluation measurements. The TAR is defined as the frequency in which the system correctly verifies a true claim of authorization. The FRR is the frequency of the system rejecting a true claim of authorization. The FAR is defined as the frequency of the system incorrectly accepting an unauthorized user [38, 44-46]. The EER is defined as the error rate when FNMR(t) and FMR(t) are equal to the threshold value t [44]. FNMR(t) and FMR(t) are defined as the distribution of false non-match rate (FNMR) and false match rate (FMR) with threshold value t varying from 0.0 to 1.0. FNMR is the fraction of genuine comparisons (a user is compared with gallery images of her / himself) that produce a matching score of less than threshold value t. FMR is the fraction of impostor comparisons (a user is compared against gallery images of all other users) that produce a matching score greater than or equal to threshold value t [45, 46].

-------------------------------------------------- Omitted --------------------------------------------------------------

To evaluate the overall performance of the proposed system, facial images for registration (as a gallery dataset) and videos for testing (as a probe dataset) were collected in an indoor environment, as illustrated in Figure 11. The newly collected dataset consists of ten subjects with ten still images and one video sequence of each subject. The still images were captured at a two-meter distance from the system camera, and the videos were recorded while the subject was walking toward the system from a five-meter distance. The videos were recorded a few days after the still images were captured. Ten still images per subject were used as the gallery dataset, and twenty image frames per video were used as the probe dataset.

-------------------------------------------------- Omitted --------------------------------------------------------------

4.2. Experimental Eavluations

To show the face recognition performance of the proposed method, we compare our method with Gabor-LBP histogram. Table 1 shows that the proposed face recognition approach achieves better recognition accuracy with smaller facial feature vectors compared with the Gabor-LBP histogram-based approach. As shown in Figure 10(a), the E-face dataset contains complicated backgrounds and variations of facial image conditions such as illumination changes, small variations of pose and facial expression, and the presence or absence of spectacles. This experiment indicates that the proposed approach is more invariant to complicated backgrounds and variations of facial image conditions than the Gabor-LBP histogram-based approach.

-------------------------------------------------- Omitted --------------------------------------------------------------

We summarize the quantitative and qualitative analysis with existing methodologies in Table 5. The deep learning based approach [26] shows superior performance in accuracy over the other approaches. To apply deep learning based approach to the embedded system, specialized neural computing device such as the Intel Movidius Neural Compute Stick is needed. Many systems are implemented using Raspberry Pi and employ handcrafted features.

Table 5. Comparison to the standalone access control systems based on facial image recognition.

Criteria

Saraf et al. [25]

Boka et al. [26]

Bakshsi et al. [27]

Sagar et al. [28]

Sajjad et al. [29]

Proposed Method

Embedded H/W

Raspberry Pi

Raspberry Pi with the Intel Movidius Neural Compute Stick

Arduino UNO

Renesas board (RL78 G13)

Raspberry Pi 3 model B, ARM Cortex-A53, 1.2 GHz processor, Video Core IV GPU

Cortex dual core CPU, 2.50 DMIPS/MHz per core

Camera

Single Camera

Two Camera

Single Camera

Single Camera

Raspberry Pi Camera

Single Camera

Face Detection

Haar cascade method

Haar-feature based approach

N/A

PCA, LDA, Histogram Equalization

Viola Jones Method

AdaBoost with LBP

Facial Feature

LBP histogram

FaceNet

Global feature with PCA

PCA, LDA, Histogram Equalization

Bag of Words with Oriented FAST and Rotated BRIEF

Gaussian Derivative Filter-LBP histogram

Face Identification

N/A

k-NN

Euclidean Distance

N/A

SVM

Histogram Intersection

Execution Time

N/A

N/A

N/A

10 sec. for authentication

9 sec.

First 10 matches

190ms

Per image

Overall Accuracy

N/A

99.60% accuracy on LFW dataset

N/A

90.00% accuracy on own dataset

91.33% average accuracy on

Face-95 dataset

95.00% true accept rate on own video stream dataset

Reviewer 3 Report

The paper is interesting and the overall contribution is sufficient for its publication. 

Some minor notes are here following.

The state of the art needs a review going to search for other algorithms of face recognition and detection in real-time and trying to demonstrate because their algorithm is better, for example: "Chen S., Liu Y., Gao X., Han Z. (2018) MobileFaceNets: Efficient CNNs for Accurate Real-Time Face Verification on Mobile Devices. In: Zhou J. et al. (eds) Biometric Recognition. CCBR 2018. Lecture Notes in Computer Science, vol 10996. Springer, Cham "

At row 105:  "In Section 3, we provide the related works.", but the right section is section 2.

At row 206 and row 306: "A user does not need to touch the RFID reader to gain access right because the system reads user ID from active RFID card at a distance when the proximity sensor detects a user."  The sentences are repeated, suggest to delete this repetition. 

Author Response

Comments and Suggestions for Authors

The paper is interesting and the overall contribution is sufficient for its publication. 

Some minor notes are here following.

The state of the art needs a review going to search for other algorithms of face recognition and detection in real-time and trying to demonstrate because their algorithm is better, for example: "Chen S., Liu Y., Gao X., Han Z. (2018) MobileFaceNets: Efficient CNNs for Accurate Real-Time Face Verification on Mobile Devices. In: Zhou J. et al. (eds) Biometric Recognition. CCBR 2018. Lecture Notes in Computer Science, vol 10996. Springer, Cham "

â–º Response: Thank you for your kind comment. Yes, we added recent research works to the references, and cited the paper in Related Works section as follows:

Deep learning approach, in particular Convolutional Neural Network (CNN), has been widely applied to the facial image recognition applications. Zhang et al. [35] propose a deep cascaded multitask framework for face detection and alignment under unconstrained environment. The proposed approach exploits the inherent correlation between detection and alignment, and also leverages a cascaded architecture with three stages of deep convolutional networks to predict face and landmark location in a coarse-to-fine manner. Schroff et al. [36] present a unified system, called FaceNet, for face verification, recognition and clustering, which is based on learning a Euclidean embedding per image using deep convolutional neural network. With FaceNet embedding as feature vector, face verification, recognition and clustering can be easily solved using standard techniques such as thresholding, k-NN classification, and k-means. Chen et al. [37] propose an efficient CNN models, called MobileFaceNets, for accurate real-time face verification on mobile devices. The proposed method use no more than 1 million parameters.

K. Zhang, Z. Zhang, Z. Li, Y. Qiao, “Joint Face Detection and Alignment using Multi-task Cascaded Convolutional Networks,” IEE Signal Processing Letters, vol. 23, issue 23, pp. 1499-1503, 2016. F. Schroff, D. Kalenichenko, and J. Philbin, “FaceNet: A unified embedding for face recognition and clustering,” in Proc. CVPR, Boston, MA, USA, June 2015. S. Chen, Y. Liu, X. Gao, and Z. Han. (2018) MobileFaceNets: Efficient CNNs for Accurate Real-Time Face Verification on Mobile Devices. In: Zhou J. et al. (eds) Biometric Recognition. CCBR 2018. Lecture Notes in Computer Science, vol 10996. Springer, Cham

At row 105:  "In Section 3, we provide the related works.", but the right section is section 2.

â–º Response: Thank you for your kind comment. We corrected the sentence as follows:

The remaining parts of this paper are organized as follows. In Section 2, we provide the related works.

At row 206 and row 306: "A user does not need to touch the RFID reader to gain access right because the system reads user ID from active RFID card at a distance when the proximity sensor detects a user."  The sentences are repeated, suggest to delete this repetition. 

â–º Response: Thank you for your kind comment. We removed duplicated sentence in Section “4. Experimental Results”.

We thank reviewers for detailed and useful comments again. We also deeply thank you for contributing to enhancing the completion of this paper by your constructive criticism. We, authors did our utmost to complete this paper. The revised manuscript has been resubmitted to your journal.

Thank you.

Round 2

Reviewer 1 Report

This article presents a face recognition system based on handcrafted features. The authors use a combination of LBP, LDA, and Adaboost for face detection and a combination of Gaussian derivatives and LBP for face description. They focus on handcrafted approaches due to their low computational cost, which is an important feature in embedded systems.

*** Technical comments ***

There are different technical issues in this work:

1) Face detection and description approaches are not adequately described. One may not reproduce these methods by their description. Although the authors mention that they use the same detection method presented in their previous works [22], both articles fail to provide a detailed explanation. For instance, the following questions must be answered in the text:

Line 189: Are you using uniform LBP for face detection? Figure 1 gives this impression, but no details are given in the text. Line 190: How many images are used in the "multi-level pyramid images"? What are their sizes? What is the face size? Are these values the same for faces and eyes? Line 197: How many weak classifiers were used to create the final classifier? Line 202: What is the face size? How the face is cropped based on the eye positions? Line 252: How many Gaussian derivatives are being used? What are their parameters? Line 271: Are you using uniform LBP for face description as well?

2) There is not a major technical difference between this work and the authors' previous work [22]. Although their purpose is different (the previous addresses age and gender estimation and this one identity verification), both use the same combination of approaches for face detection and description. What is the novelty of this work?

3) The title gives the idea that the system works "at a distance", but the authors later say that identity is only verified "when a user is close to the system" (Line 296). The authors should explicitly define a range of operation distance.

4) The definition of EER is incorrect.

5) Why are the authors showing identification results (Rank-1, CMC curves) for E-face and XM2VTS databases? This work is focused on face verification due to the use of RFID, so the results should be presented in terms of FAR, FRR, ROC curves and so on, as was done for the PF07 database.

6) In Table 3, FRR and TAR values are incompatible (FRR = 100 - TAR).

7) None of the databases used in this work allow a direct comparison in terms of accuracy to other works listed in Table 5. The authors should consider using the same databases used in these works as well.

*** Comments on article organization ***

A few adjustments would improve the overall organization of this work:

* Do not divide figures and tables across different pages (Figure 2, Figure 9, Figure 11, Table 5)

* Line 219: Fix equation citations.

* Section 3.4 should probably appear at the beginning of Section 3, with the following sections providing details on each important part (A, B, C, and D). Figure 8 is the most informative diagram and serves as an introduction to the proposed system. Besides, Figure 6 is not required, as all of its content appears in Figure 8, and face pre-processing is not detailed in the text.

*** Comments on language usage ***

Extensive English changes are required. Below are listed some examples found in the abstract and the first paragraph of the introduction:

* Line 22: "to lack" -> "to the lack"

* Line 22: "resource" -> "resources"

* Line 22: "The lightweight" -> "lightweight"

* Line 25: "lighting / illumination" -> "lighting/illumination"

* Line 26: "presents design" -> "presents the design"

* Line 33: "on benchmark dataset" -> "on benchmark datasets"

* Line 34: "achieves 91.5%" -> "achieves a 91.5%"

* Line 36: "under indoor" -> "under an indoor"

* Line 41: "biometric" -> "biometrics"

* Line 44: "biometric based" -> "biometric-based"

* Line 44: "fingerprint and iris based" -> "fingerprint- and iris-based"

* Line 46: "demands active" -> "demands the active"

* Line 47: "fingerprint and iris based" -> "fingerprint- and iris-based"

* Line 51: "proposed automatic" -> "proposed an automatic"

* Line 52: "discrimnant" -> "discriminant"

Reviewer 2 Report

Authors have properly replied to the review comments improving the scientific quality of the paper

Author Response

With respect, Dear Reviewer,

To make the contents a more scientifically sound one, the inadequate expressions have been revised (red color) with the help of an English native speaker. Also, the readability of the manuscript has been improved with an assistance of a native speaker.